# GraphZoom: A Multi-level Spectral Approach for Accurate and Scalable Graph Embedding

**Chenhui Deng***
Cornell University, Ithaca, USA
cd574@cornell.edu

**Zhiqiang Zhao***
Michigan Technological University, Houghton, USA
qzzhao@mtu.edu

**Yongyu Wang**
Michigan Technological University, Houghton, USA
yongyuw@mtu.edu

**Zhiru Zhang**
Cornell University, Ithaca, USA
zhiruz@cornell.edu

**Zhuo Feng**
Stevens Institute of Technology, Hoboken, USA
zfeng12@stevens.edu

## Abstract

Graph embedding techniques have been increasingly deployed in a multitude of different applications that involve learning on non-Euclidean data. However, existing graph embedding models either fail to incorporate node attribute information during training or suffer from node attribute noise, which compromises the accuracy. Moreover, very few of them scale to large graphs due to their high computational complexity and memory usage. In this paper we propose GraphZoom, a multi-level framework for improving both accuracy and scalability of unsupervised graph embedding algorithms.[1] GraphZoom first performs graph fusion to generate a new graph that effectively encodes the topology of the original graph and the node attribute information. This fused graph is then repeatedly coarsened into much smaller graphs by merging nodes with high spectral similarities. GraphZoom allows any existing embedding methods to be applied to the coarsened graph, before it progressively refine the embeddings obtained at the coarsest level to increasingly finer graphs. We have evaluated our approach on a number of popular graph datasets for both transductive and inductive tasks. Our experiments show that GraphZoom can substantially increase the classification accuracy and significantly accelerate the entire graph embedding process by up to $40.8\times$, when compared to the state-of-the-art unsupervised embedding methods.

## 1 Introduction

Recent years have seen a surge of interest in graph embedding, which aims to encode nodes, edges, or (sub)graphs into low dimensional vectors that maximally preserve graph structural information. Graph embedding techniques have shown promising results for various applications such as vertex classification, link prediction, and community detection (Zhou et al., 2018); (Cai et al., 2018); (Goyal & Ferrara, 2018). However, current graph embedding methods have several drawbacks in terms of either accuracy or scalability. On the one hand, random-walk-based embedding algorithms, such as DeepWalk (Perozzi et al., 2014) and node2vec (Grover & Leskovec, 2016), attempt to embed a graph based on its topology without incorporating node attribute information, which limits their embedding power. Later, graph convolutional networks (GCN) are developed with the basic notion that node embeddings should be smoothed over the entire graph (Kipf & Welling, 2016). While GCN leverages both topology and node attribute information for simplified graph convolution in each layer, it may suffer from high-frequency noise in the initial node features, which compromises

---

*Equal contributions
[1]Source code of GraphZoom is freely available at: github.com/cornell-zhang/GraphZoom.

the embedding quality (Maehara, 2019). On the other hand, few embedding algorithms can scale well to large graphs with millions of nodes due to their high computation and storage cost (Zhang et al., 2018a). For example, graph neural networks (GNNs) such as GraphSAGE (Hamilton et al., 2017) collectively aggregate feature information from the neighborhood. When stacking multiple GNN layers, the final embedding vector of a node involves the computation of a large number of intermediate embeddings from its neighbors. This will not only cause drastic increase in the amount of computation among nodes, but also lead to high memory usage for storing the intermediate results.

In literature, increasing the accuracy and improving the scalability of graph embedding methods are largely viewed as two orthogonal problems. Hence most research efforts are devoted to addressing only one of the problems. For instance, Chen et al. (2018) and Fu et al. (2019) proposed multi-level methods to obtain high-quality embeddings by training unsupervised models at every level; but their techniques do not improve scalability due to the additional training overhead. Liang et al. (2018) developed a heuristic algorithm to coarsen the graph by merging nodes with similar local structures. They use GCN to refine the embedding results on the coarsened graphs, which is not only time consuming to train but also potentially degrading accuracy when the number of GCN layers increases. More recently, Akbas & Aktas (2019) proposed a similar strategy to coarsen the graph, where certain useful structural properties of the graph are preserved (e.g., local neighborhood proximity). However, this work lacks proper refinement methods to improve the embedding quality.

In this paper we propose *GraphZoom*, a multi-level spectral approach to enhancing both the quality and scalability of unsupervised graph embedding methods. More concretely, GraphZoom consists of four major kernels: (1) graph fusion, (2) spectral graph coarsening, (3) graph embedding, and (4) embedding refinement. The graph fusion kernel first converts the node feature matrix into a feature graph and then fuses it with the original topology graph. The fused graph provides richer information to the ensuing graph embedding step to achieve a higher accuracy. Spectral graph coarsening produces a series of successively coarsened graphs by merging nodes based on their spectral similarities. We show that our coarsening algorithm can effectively and efficiently retain the first few eigenvectors of the graph Laplacian matrix, which is critical for preserving the key graph structures. During the graph embedding step, any of the existing unsupervised graph embedding techniques can be applied to obtain node embeddings for the graph at the coarsest level. [2] Embedding refinement is then employed to refine the embeddings back to the original graph by applying a proper graph filter to ensure embeddings are smoothed over the graph.

We evaluate the proposed GraphZoom framework on three transductive benchmarks: Cora, Citeseer and Pubmed citation networks as well as two inductive dataset: PPI and Reddit for vertex classification task. We further test the scalability of our approach on friendster dataset, which contains 8 million nodes and 400 million edges. Our experiments show that GraphZoom can improve the classification accuracy over all baseline embedding methods for both transductive and inductive tasks. Our main technical contributions are summarized as follows:

- **GraphZoom generates high-quality embeddings.** We propose novel algorithms to encode graph structures and node attribute information in a fused graph and exploit graph filtering during refinement to remove high-frequency noise. This results in a relative increase of the embedding accuracy over the prior arts by up to $19.4\%$ while reducing the execution time by at least $2\times$.
- **GraphZoom improves scalability.** Our approach can significantly reduce the embedding run time by effectively coarsening the graph without losing the key spectral properties. Experiments show that GraphZoom can accelerate the entire embedding process by up to $40.8\times$ while producing a similar or better accuracy than state-of-the-art techniques.
- **GraphZoom is highly composable.** Our framework is agnostic to underlying graph embedding techniques. Any of the existing unsupervised embedding methods, either transductive or inductive, can be incorporated by GraphZoom in a plug-and-play manner.

## 2 RELATED WORK

There is a large and active body of research on multi-level graph embedding and graph filtering, from which GraphZoom draws inspiration to boost the performance and speed of unsupervised

---

[2]In this work, we do not attempt to preserve node label information in the coarsened graph. Nonetheless, we believe that our approach can be extended to support supervised embedding models such as GAT (Gulcehre et al., 2019) and PPNP (Klicpera et al., 2019), as briefly discussed in Section 5.

embedding methods. Due to the space limitation, we only summarize some of the recent efforts in these two areas.

**Multi-level graph embedding** attempts to coarsen the original graph into a series of smaller graphs with decreasing size where existing or new graph embedding techniques can be applied at different coarsening levels. For example, Chen et al. (2018); Lin et al. (2019) generate a hierarchy of coarsened graphs and perform embedding from the coarsest level to the original one. Fu et al. (2019) construct and embed multi-level graphs through hierarchical, and the resulting embedding vectors are concatenated to obtain the final node embeddings for the original graph. These methods, however, only focus on improving embedding quality but not the scalability. Later, Zhang et al. (2018b); Akbas & Aktas (2019) attempt to make graph embedding more scalable by only embedding on the coarsest graph. However, their methods lack proper refinement methods to generate high-quality embeddings for the original graph. Liang et al. (2018) propose MILE, which only trains the coarsest graph to obtain coarse embeddings, and leverages GCN as refinement method to improve embedding quality. However, MILE requires training a GCN model which is very time consuming for large graphs and leading to poor performance when multiple GCN layers are stacked together (Li et al., 2018). In contrast to the prior arts, GraphZoom is motivated by theoretical results in spectral graph embedding (Tang & Liu, 2011) and practically efficient so that it can improve both accuracy and scalability of the unsupervised graph embedding tasks.

**Graph filtering** are direct analogs of classical filters in signal processing field, but intended for signals defined on graphs. Shuman et al. (2013) define graph filters in both vertex and spectral domains, and apply graph filter on image denoising and reconstruction tasks. Wu et al. (2019) leverage graph filter to simplify GCN model by removing redundant computation. More recently, Maehara (2019) show the fundamental link between graph embedding and filtering by proving that GCN model implicitly exploits graph filter to remove high-frequency noise from the node feature matrix; a filter neural network (gfNN) is then proposed to derive a stronger graph filter to improve the embedding results. Li et al. (2019) further derive two generalized graph filters and apply them on graph embedding models to improve their embedding quality for various classification tasks. In GraphZoom we adopt graph filter to properly smooth the intermediate embedding results during the iterative refinement step, which is crucial for improving the quality of the final embedding.

## 3 GRAPHZOOM FRAMEWORK

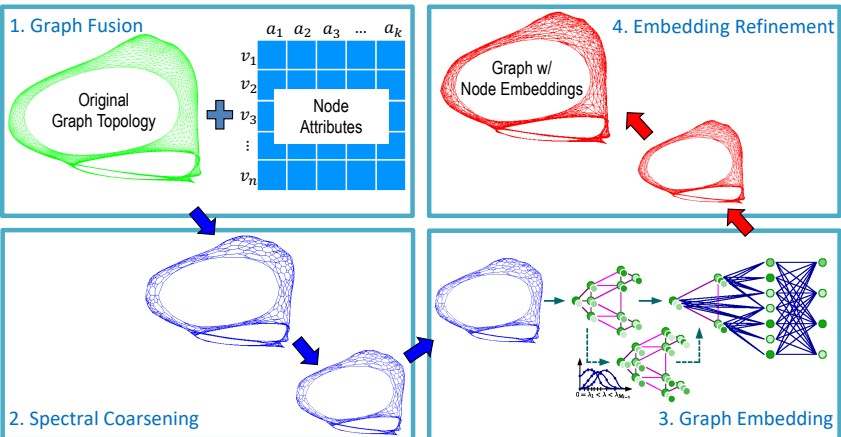

Figure 1: Overview of the GraphZoom framework.

Figure 1 shows the proposed GraphZoom framework, which consists of four key phases: Phase (1) is graph fusion, which combines the node attributes and topology information of the original graph to construct a fused weighted graph; In Phase (2), a spectral graph coarsening process is applied to form a hierarchy of coarsened fused graphs with decreasing size; In Phase (3), any of the existing graph embedding methods can be applied to the fused graph at the coarsest level; In Phase (4), the embedding vectors obtained at the coarsest level are mapped onto a finer graph using the mapping

operators determined during the coarsening phase. This is followed by a refinement (or smoothing) procedure, where the procedure in Phase (4) is applied in an iterative manner to increasingly finer graphs; Eventually the embedding vectors for the original graph are obtained. In the rest of this section, we describe each of these four phases in more detail.

## 3.1 PHASE 1: GRAPH FUSION

Graph fusion aims to construct a weighted graph that has the same number of nodes as the original graph but potentially different set of edges (weights) that encapsulate the original graph topology as well as node attribute information. Specifically, given an undirected graph $\mathcal{G} = (\mathcal{V}, \mathcal{E})$ with $N = |\mathcal{V}|$ nodes, its adjacency matrix $A_{topo} \in \mathbb{R}^{N \times N}$ and its node attribute (feature) matrix $X \in \mathbb{R}^{N \times K}$, where $K$ corresponds to the dimension of node attribute vector, graph fusion can be interpreted as a function $f(\cdot)$ that outputs a weighted graph $\mathcal{G}_{fusion} = (\mathcal{V}, \mathcal{E}_{fusion})$ represented by its adjacency matrix $A_{fusion} \in \mathbb{R}^{N \times N}$, namely, $A_{fusion} = f(A_{topo}, X)$.

We first convert the initial attribute matrix $X$ into a weighted node attribute graph $\mathcal{G}_{feat} = (\mathcal{V}, \mathcal{E}_{feat})$ by generating a k-nearest-neighbor (kNN) graph based on the $l^2\text{-}norm$ distance between the attribute vectors of each node pair. A straightforward implementation requires comparing all possible node pairs and then selecting top-$k$ nearest neighbors. However, such a naïve approach has a worst-case time complexity of $O(N^2)$, which certainly does not scale to large graphs. Our goal is to construct the attribute graph in nearly linear time by leveraging an efficient spectral graph coarsening scheme that is described in more detail in Section 3.2. More specifically, our approach starts with coarsening the original graph $\mathcal{G}$ to obtain a substantially reduced graph that has much fewer nodes with an $O(|\mathcal{E}|)$ time complexity. Note that this procedure bears similarity to spectral graph clustering, which aims to group nodes into clusters of high conductance (Peng et al., 2015). Once the node clusters are formed through spectral coarsening, we can select the top-$k$ nearest neighbors within each cluster with $O(M^2)$ comparisons, where $M$ is the average node count per cluster. Since we have roughly $N/M$ clusters, the time complexity for constructing the approximate kNN graph is $O(MN)$. When a proper coarsening ratio is chosen where $M \ll N$ (e.g., $M = 50$), the overall time complexity will become almost linear.

After the attribute graph is formed, we assign a weight to each edge based on the cosine similarity between the attribute vectors of the two incident nodes, namely $w_{i,j} = (X_{i,:} \cdot X_{j,:})/(\|X_{i,:}\|\|X_{j,:}\|)$, where $X_{i,:}$ and $X_{j,:}$ are the attribute vectors of nodes $i$ and $j$. Finally, we can construct the fused graph by combining the topology graph and the attribute graph using a weighted sum: $A_{fusion} = A_{topo} + \beta A_{feat}$, where $\beta$ allows us to balance the topological and node attribute information during the fusion process. This fused graph can be fed into any downstream graph embedding procedures.

## 3.2 PHASE 2: SPECTRAL COARSENING

**Graph coarsening via global spectral embedding.** To reduce the size of the original graph while preserving important spectral properties (e.g., the first few eigenvalues and eigenvectors of the graph Laplacian matrix [3]), a straightforward way is to first embed the graph into a $k$-dimensional space using the first $k$ eigenvectors of the graph Laplacian matrix, which is also known as the spectral graph embedding technique (Belkin & Niyogi, 2003; Peng et al., 2015). Next, the graph nodes that are close to each other in the low-dimensional embedding space can be aggregated to form the coarse-level nodes and subsequently the reduced graph. However, it is very costly to calculate the eigenvectors of the original graph Laplacian, especially for very large graphs.

**Graph coarsening via local spectral embedding.** In this work, we leverage an efficient yet effective local spectral embedding scheme to identify node clusters based on emerging graph signal processing techniques (Shuman et al., 2013). There are obvious analogies between the traditional signal processing (Fourier analysis) and graph signal processing: (1) The signals at different time points in classical Fourier analysis correspond to the signals at different nodes in an undirected graph; (2) The more slowly oscillating functions in time domain correspond to the graph Laplacian eigenvectors associated with lower eigenvalues or the more slowly varying (smoother) components across the graph. Instead of directly using the first few eigenvectors of the original graph Laplacian,

---

[3]Laplacian matrix $L$ is defined as $L = D - A$, where $D$ is degree matrix and $A$ is adjacency matrix.

we apply the simple smoothing (low-pass graph filtering) function to $k$ random vectors to obtain smoothed vectors for $k$-dimensional graph embedding, which can be achieved in linear time.

Consider a random vector (graph signal) $x$ that is expressed with a linear combination of eigenvectors $\boldsymbol{u}$ of the graph Laplacian. We adopt low-pass graph filters to quickly filter out the high-frequency components of the random graph signal or the eigenvectors corresponding to high eigenvalues of the graph Laplacian. By applying the smoothing function on $\boldsymbol{x}$, we obtain a smoothed vector $\tilde{\boldsymbol{x}}$, which is basically the linear combination of the first few eigenvectors:

$$\boldsymbol{x} = \Sigma_{i=1}^{N}\alpha_i\boldsymbol{u}_i \quad \xRightarrow{\text{smoothing}} \quad \tilde{\boldsymbol{x}} = \Sigma_{i=1}^{n}\tilde{\alpha}_i\boldsymbol{u}_i , \;\; n \ll N \tag{1}$$

More specifically, we apply a few (typically five to ten) Gauss-Seidel iterations for solving the linear system of equations $L_{\mathcal{G}}x^{(i)} = 0$ to a set of $t$ initial random vectors $T = (x^{(1)}, \ldots, x^{(t)})$ that are orthogonal to the all-one vector $\mathbf{1}$ satisfying $\mathbf{1}^{\top}x^{(i)} = 0$, and $L_{\mathcal{G}}$ is the Laplacian matrix of graph $\mathcal{G}$ or $\mathcal{G}_{fusion}$.

Based on the smoothed vectors in $T$, we embed each node into a $t$-dimensional space such that nodes $p$ and $q$ are considered spectrally similar if their low-dimensional embedding vectors $x_p \in \mathbb{R}^t$ and $x_q \in \mathbb{R}^t$ are highly correlated. Here the node distance is measured by the spectral node affinity $a_{p,q}$ for neighboring nodes $p$ and $q$ (Livne & Brandt, 2012; Chen & Safro, 2011):

$$a_{p,q} = \frac{|(\boldsymbol{T}_{p,:}, \boldsymbol{T}_{q,:})|^2}{\|\boldsymbol{T}_{p,:}\|^2\|\boldsymbol{T}_{q,:}\|^2}, \quad (\boldsymbol{T}_{p,:}, \boldsymbol{T}_{q,:}) = \Sigma_{k=1}^{t}(\boldsymbol{x}_p^{(k)} \cdot \boldsymbol{x}_q^{(k)}) \tag{2}$$

Once the node aggregation schemes are determined, we can easily obtain the graph mapping operator $H_i^{i+1}$ between two coarsening levels $i$ and $i+1$. More precisely, $H_i^{i+1}$ is a matrix of size $|\mathcal{V}_{\mathcal{G}_{i+1}}| \times |\mathcal{V}_{\mathcal{G}_i}|$. $(H_i^{i+1})_{p,q} = 1$ if node $q$ in graph $\mathcal{G}_i$ is aggregated to (clustered) node $p$ in graph $\mathcal{G}_{i+1}$; otherwise, it is set to 0. Additional discussions on the properties of $H_i^{i+1}$ are available in Appendix F, which shows this operator a surjective mapping and locality preserving.

We leverage $H_0^1, H_1^2, \cdots, H_{l-1}^l$ for constructing a series of spectrally-reduced graphs $\mathcal{G}_1, \mathcal{G}_2, \cdots, \mathcal{G}_l$. Specifically, The coarser graph Laplacian $L_{\mathcal{G}_{i+1}}$ can be computed by Eq. (3). It is worth noting that $\mathcal{G}_0$ (i.e., $\mathcal{G}_{fusion}$) is the original fused graph and $|\mathcal{V}_0| = N > |\mathcal{V}_1| > \cdots > |\mathcal{V}_l|$.

$$\boldsymbol{L}_{\mathcal{G}_{i+1}} = \boldsymbol{H}_i^{i+1}\boldsymbol{L}_{\mathcal{G}_i}\boldsymbol{H}_{i+1}^i, \;\; \boldsymbol{H}_{i+1}^i = (\boldsymbol{H}_i^{i+1})^T \tag{3}$$

We emphasize that the aggregation scheme based on the above spectral node affinity calculations will have a (linear) complexity of $O(|\mathcal{E}_{fusion}|)$ and thus allow preserving the spectral (global or structural) properties of the original graph in a highly efficient and effective way. As suggested in (Zhao & Feng, 2019; Loukas, 2019), a spectral sparsification procedure can be applied to effectively control densities of coarse level graphs. In this work, a similarity-aware spectral sparsification tool "GRASS" (Feng, 2018) has been adopted for achieving a desired graph sparsity at the coarsest level.

### 3.3 PHASE 3: GRAPH EMBEDDING

**Embedding the Coarsest Graph.** Once the coarsest graph $\mathcal{G}_l$ is constructed, node embeddings $E_l$ on $\mathcal{G}_l$ can be obtained by $E_l = g(\mathcal{G}_l)$, where $g(\cdot)$ can be any unsupervised embedding methods.

### 3.4 PHASE 4: EMBEDDING REFINEMENT

Given $\boldsymbol{E}_{i+1}$, the node embeddings of graph $\mathcal{G}_{i+1}$ at level $i+1$, we can use the corresponding projection operator $H_{i+1}^i$ to project $\boldsymbol{E}_{i+1}$ to $\mathcal{G}_i$, which is the finer graph at level $i$:

$$\hat{\boldsymbol{E}}_i = \boldsymbol{H}_{i+1}^i\boldsymbol{E}_{i+1} \tag{4}$$

Due to the property of the projection operator, embedding of the node in the coarser graph will be directly copied to the nodes of the same aggregation set in the finer graph at the preceding level. In this case, spectrally-similar nodes in the finer graph will have the same embedding results if they are aggregated into a single node during the coarsening phase.

To further improve the quality of the mapped embeddings, we apply a local refinement process motivated by Tikhonov regularization to smooth the node embeddings over the graph by minimizing the following objective:

$$\min_{\boldsymbol{E}_i}\{\left\|\boldsymbol{E}_i - \hat{\boldsymbol{E}}_i\right\|_2^2 + tr(\boldsymbol{E}_i^{\mathsf{T}}\boldsymbol{L}_i\boldsymbol{E}_i)\} \tag{5}$$

where $L_i$ and $E_i$ are the normalized Laplacian matrix and mapped embedding matrix of the graph at the $i$-th coarsening level, respectively. We obtain the refined embedding matrix $\tilde{\boldsymbol{E}}_i$ by solving Eq. (5). Here the first term enforces the refined embeddings to agree with mapped embeddings, while the second term employs Laplacian smoothing to smooth $\tilde{\boldsymbol{E}}_i$ over the graph. By taking the derivative of the objective function in Eq. (5) and setting it to zero, we have:

$$\boldsymbol{E}_i = (\boldsymbol{I} + \boldsymbol{L}_i)^{-1}\hat{\boldsymbol{E}}_i \tag{6}$$

where $\boldsymbol{I}$ is the identity matrix. However, obtaining refined embeddings in this manner can be very inefficient since it involves matrix inversion whose time complexity is $O(N^3)$. Instead, we exploit a more efficient spectral graph filter to smooth the embeddings.

By transforming $h(L) = (I + L)^{-1}$ into spectral domain, we obtain the graph filter: $h(\lambda) = (1 + \lambda)^{-1}$. To avoid the inversion term, we approximate $h(\lambda)$ by its first-order Taylor expansion, namely, $\tilde{h}(\lambda) = 1 - \lambda$. We then generalize $\tilde{h}(\lambda)$ to $\tilde{h}_k(\lambda) = (1 - \lambda)^k$, where k controls the power of graph filter. After transforming $\tilde{h}_k(\lambda)$ into the spatial domain, we have: $\tilde{h}_k(L) = (I - L)^k = (D^{-\frac{1}{2}}AD^{-\frac{1}{2}})^k$, where $A$ is the adjacency matrix and $D$ is the degree matrix. It is not difficult to show that adding a proper self-loop for every node in the graph will allow $\tilde{h}_k(L)$ to more effectively filter out high-frequency noise components (Maehara, 2019) (see Appendix H). Thus, we modify the adjacency matrix as $\tilde{A} = A + \sigma I$, where $\sigma$ is a small value to ensure every node has its own self-loop. Finally, the low-pass graph filter can be utilized to smooth the mapped embedding matrix, as shown in (7):

$$\boldsymbol{E}_i = (\tilde{\boldsymbol{D}}_i^{-\frac{1}{2}}\tilde{\boldsymbol{A}}_i\tilde{\boldsymbol{D}}_i^{-\frac{1}{2}})^k\hat{\boldsymbol{E}}_i = (\tilde{\boldsymbol{D}}_i^{-\frac{1}{2}}\tilde{\boldsymbol{A}}_i\tilde{\boldsymbol{D}}_i^{-\frac{1}{2}})^k\boldsymbol{H}_{i+1}^i\boldsymbol{E}_{i+1} \tag{7}$$

We iteratively apply Eq. (7) to obtain the embeddings of the original graph (i.e., $E_0$). Note that our refinement stage does not involve training and can be simply considered as several (sparse) matrix multiplications, which can be efficiently computed.

## 4 EXPERIMENTS

We have performed comparative evaluation of GraphZoom framework against several state-of-the-art unsupervised graph embedding techniques and multi-level embedding frameworks on five standard graph-based datasets (transductive as well as inductive). In addition, we evaluate the scalability of GraphZoom on Friendster dataset, which contains 8 million nodes and 400 million edges. Finally, we conduct ablation study to understand the effectiveness of the major GraphZoom kernels.

### 4.1 EXPERIMENTAL SETUP

Table 1: Statistics of datasets used in our experiments.

| Dataset | Type | Task | Nodes | Edges | Classes | Features |
|---|---|---|---|---|---|---|
| Cora | Citation network | Transductive | 2,708 | 5,429 | 7 | 1,433 |
| Citeseer | Citation network | Transductive | 3,327 | 4,732 | 6 | 3,703 |
| Pubmed | Citation network | Transductive | 19,717 | 44,338 | 3 | 500 |
| PPI | Biology network | Inductive | 14,755 | 222,055 | 121 | 50 |
| Reddit | Social network | Inductive | 232,965 | 57,307,946 | 210 | 5,414 |
| Friendster | Social network | Transductive | 7,944,949 | 446,673,688 | 5,000 | N/A |

**Datasets.** Table 1 reports the statistics of the datasets used in our experiments. We include Cora, Citeseer, Pubmed, and Friendster for evaluation on transductive learning tasks, and PPI as well as

Reddit for inductive learning. We split the training and testing data in the same way as suggested in Kipf & Welling (2016); Hamilton et al. (2017).

**Transductive baseline models.** A number of popular graph embedding techniques are transductive learning methods, which require all nodes in the graph be present during training. Hence such embedding models must be retrained whenever a new node is added. Here we compare GraphZoom with three transductive models: DeepWalk (Perozzi et al., 2014), node2vec (Grover & Leskovec, 2016), and Deep Graph Infomax (DGI) (Veličković et al., 2019) [4]. These methods have shown state-of-the-art unsupervised embedding results on the datasets used in our experiments. In addition, we compare GraphZoom with two multi-level graph embedding frameworks: HARP (Chen et al., 2018) and MILE (Liang et al., 2018), which have reported improvement over DeepWalk and node2vec in either embedding quality or scalability.

**Inductive baseline models.** In contrast to transductive tasks, training an inductive graph embedding model does not require seeing the whole graph structure. Hence the resulting trained model can still be applied when new nodes are added to graph. To show GraphZoom can also enhance inductive learning, we compare it against GraphSAGE (Hamilton et al., 2017) using four different aggregation functions including GCN, mean, LSTM, and pooling.

More details of datasets and baselines are available in Appendix A and B. We optimize hyperparameters of DeepWalk, node2vec, DGI, and GraphSAGE to achieve highest possible accuracy on original datasets; we then choose the same hyperparameters to embed the coarsened graph in HARP, MILE, and GraphZoom. We run all the experiments on a Linux machine with an Intel Xeon Gold 6242 CPU (32 cores @ 2.40GHz) and 384 GB of RAM.

## 4.2 PERFORMANCE AND SCALABILITY OF GRAPHZOOM

Since HARP and MILE only support transductive learning, we compare them with GraphZoom using DeepWalk, node2vec, and DGI (Veličković et al., 2019) as embedding kernels. For inductive tasks, we compare GraphZoom with GraphSAGE using four different aggregation functions.

Tables 2 and 3 report the mean classification accuracy for the transductive task and micro-averaged F1 score for the inductive task, respectively, as well as the execution time for all baselines and Graph-Zoom. Specifically, we use the CPU time for graph embedding as the execution time of DeepWalk, node2vec, DGI, and GraphSAGE. We further measure the execution time of HARP and MILE by summing up CPU time for graph coarsening, graph embedding, and embedding refinement. Similarly, we add up the CPU time for graph fusion, graph coarsening, graph embedding, and embedding refinement as the execution time of GraphZoom. In regard to hyperparameters, we use 10 walks with a walk length of 80, a window size of 10, and an embedding dimension of 128 for both DeepWalk and node2vec; we further set the return parameter $p$ and the in-out parameter $q$ in node2vec as 1.0 and 0.5, respectively. Moreover, we choose early stopping strategy for DGI with a learning rate of 0.001 and an embedding dimension of 512. Apropos the configuration of GraphSAGE, we train a two-layer model for one epoch, with a learning rate of 0.00001, an embedding dimension of 128, and a batch size of 256.

**Comparing GraphZoom with baseline embedding methods.** We show the results of GraphZoom with three coarsening levels for transductive learning and two levels for inductive learning. The size of coarsened graphs and results with larger coarsening level are available in Appendix D, Figure 3 (blue curve), and the Appendix J. Our results demonstrate that GraphZoom is agnostic to underlying embedding methods and capable of boosting the accuracy and speed of state-of-the-art unsupervised embedding methods on various datasets.

More specifically, for transductive learning tasks, GraphZoom improves classification accuracy upon both DeepWalk and node2vec by a relative gain of 8.3%, 10.4%, and 19.4% [5] on Cora, Pubmed, and Citeseer, respectively, while achieving up to $40.8\times$ run-time reduction. In regard to comparing with DGI, GraphZoom achieves comparable or better accuracy with speedup up to $11.2\times$. Similarly, GraphZoom outperforms all the baselines by a margin of 3.4% and 3.3% on PPI and Reddit for inductive learning tasks, respectively, with speedup up to $7.6\times$. These results indicate that our

---

[4]DGI also supports inductive tasks, although the authors have only released the source code for transductive learning at github.com/PetarV-/DGI, as of the date of this experiment.

[5]GZoom(N2V, $l$=1) improves over node2vec on Citeseer by $19.4\% = (54.7 - 45.8)/45.8$.

Table 2: Summary of results in terms of mean classification accuracy and CPU time for transductive tasks, on Cora, Citeseer, and Pubmed datasets — DW, N2V, and GZoom denote DeepWalk, node2vec, and GraphZoom, respectively; $l$ means the graph coarsening level; GZoom_F+MILE represents the best performance achieved when adding GraphZoom fusion kernel into MILE.

| Method | Cora | | Citeseer | | Pubmed | |
|---|---|---|---|---|---|---|
| | Accuracy(%) | Time(secs) | Accuracy(%) | Time(secs) | Accuracy(%) | Time(mins) |
| DeepWalk | 71.4 | 97.8 | 47.0 | 120.0 | 69.9 | 14.1 |
| HARP(DW) | 71.3 | 296.7 (0.3×) | 43.2 | 272.4 (0.4×) | 70.6 | 33.9 (0.4×) |
| MILE(DW, $l$=1) | 71.9 | 68.7 (1.4×) | 46.5 | 53.7 (2.2×) | 69.6 | 7.0 (2.0×) |
| MILE(DW, $l$=2) | 71.3 | 30.9 (3.2×) | 47.3 | 22.5 (5.3×) | 66.7 | 4.4 (2.3×) |
| MILE(DW, $l$=3) | 70.6 | 15.9 (6.1×) | 47.1 | 9.9 (12.1×) | 64.5 | 2.5 (5.8×) |
| GZoom_F+MILE(DW) | 73.8 | 70.6 (1.4×) | 48.9 | 24.7 (4.9×) | 72.1 | 7.0 (2.0×) |
| **GZoom**(DW, $l$=1) | 76.9 | 39.6 (2.5×) | 49.7 | 19.6 (2.1×) | 75.3 | 4.0 (3.6×) |
| **GZoom**(DW, $l$=2) | **77.3** | 15.6 (6.3×) | **50.8** | 6.7 (6.0×) | 75.9 | 1.7 (8.3×) |
| **GZoom**(DW, $l$=3) | 75.1 | **2.4 (40.8×)** | 49.5 | **1.3 (30.8×)** | **77.2** | **0.6 (23.5×)** |
| node2vec | 71.5 | 119.7 | 45.8 | 126.9 | 71.3 | 15.6 |
| HARP(N2V) | 72.3 | 171.0 (0.7×) | 44.8 | 174.3 (0.7×) | 70.1 | 46.1 (0.3×) |
| MILE(N2V, $l$=1) | 72.1 | 57.3 (2.1×) | 46.1 | 60.9 (2.1×) | 70.8 | 7.3 (2.1×) |
| MILE(N2V, $l$=2) | 71.8 | 30.0 (4.0×) | 45.7 | 28.8 (4.4×) | 67.3 | 4.3 (3.6×) |
| MILE(N2V, $l$=3) | 68.5 | 16.5 (7.2×) | 45.2 | 15.6 (8.1×) | 61.8 | 1.8 (8.0×) |
| GZoom_F+MILE(N2V) | 74.3 | 59.2 (2.0×) | 48.3 | 62.3 (2.0×) | 72.9 | 7.3 (2.1×) |
| **GZoom**(N2V, $l$=1) | **77.3** | 43.5 (2.8×) | **54.7** | 38.1 (3.3×) | 77.0 | 3.0 (5.2×) |
| **GZoom**(N2V, $l$=2) | 77.0 | 13.5 (8.9×) | 51.7 | 15.3 (8.3×) | **77.8** | 1.5 (10.4×) |
| **GZoom**(N2V, $l$=3) | 75.3 | **3.0 (39.9×)** | 50.7 | **4.5 (28.2×)** | 77.4 | **0.4 (39.0×)** |
| DGI | 82.3 | 89.7 | **71.8** | 94.6 | 76.8 | 23.7 |
| MILE(DGI, $l$=1) | 80.9 | 45.9 (1.9×) | 69.9 | 53.2 (1.8×) | 76.1 | 10.4 (2.3×) |
| MILE(DGI, $l$=2) | 80.3 | 27.5 (3.3×) | 69.2 | 31.1 (3.0×) | 74.3 | 4.3 (5.5×) |
| MILE(DGI, $l$=3) | 79.2 | 15.6 (4.4×) | 67.9 | 18.5 (5.1×) | 74.4 | 2.7 (8.8×) |
| GZoom_F+MILE(DGI) | 81.3 | 45.9 (1.9×) | 70.4 | 52.3 (1.8×) | 75.9 | 10.4 (2.3×) |
| **GZoom**(DGI, $l$=1) | **83.9** | 27.3 (3.3×) | 71.1 | 29.3 (3.2×) | 77.1 | 9.6 (2.5×) |
| **GZoom**(DGI, $l$=2) | 83.8 | 15.2 (5.9×) | 70.8 | 17.9 (5.3×) | **77.6** | 4.4 (5.4×) |
| **GZoom**(DGI, $l$=3) | 83.5 | **8.0 (11.2×)** | 70.7 | **9.6 (9.8×)** | 76.9 | **2.1 (11.2×)** |

multi-level spectral approach improves both embedding speed and quality — GraphZoom runs much faster since we only train the embedding model on the smallest fused graph at the coarsest level. In addition to reducing the graph size, our coarsening method further filters out redundant information from the original graph while preserving key spectral properties for the embedding. Hence we also observe improved embedding quality in terms of classification accuracy.

**Comparing GraphZoom with multi-level frameworks.** As shown in Table 2, HARP slightly improves the accuracy in several cases but increases the CPU time. Although MILE improves both accuracy and speed over a few baseline embedding methods, the performance of MILE becomes worse with increasing coarsening levels. For instance, the classification accuracy of MILE drops from 0.708 to 0.618 on Pubmed dataset with node2vec as the embedding kernel. GraphZoom achieves a better accuracy and speedup compared to MILE with the same coarsening level across all datasets. Moreover, when increasing the coarsening levels, namely, decreasing number of nodes on the coarsened graph, GraphZoom still produces comparable or even a better embedding accuracy with much shorter CPU times. This further confirms GraphZoom can retain the key graph structure information to be utilized by underlying embedding models to boost embedding quality. More results of GraphZoom on non-attributed graph for both node classification and link prediction tasks are available in Appendix K.

**GraphZoom for large graph embedding.** To show GraphZoom can significantly improve performance and scalability of underlying embedding model on large graph, we test GraphZoom and MILE on Friendster dataset, which contains 8 million nodes and 400 million edges. For both cases,

Table 3: Summary of results in terms of micro-averaged F1 score and CPU time for inductive tasks, on PPI and Reddit datasets — The baselines are GraphSAGE with four different aggregation functions. GZoom and GSAGE denote GraphZoom and GraphSAGE, respectively; $l$ means the graph coarsening level.

| Method | PPI | | Reddit | |
|---|---|---|---|---|
| | Micro-F1 | Time(mins) | Micro-F1 | Time(hours) |
| GraphSAGE-GCN | 0.601 | 9.6 | 0.908 | 10.1 |
| **GZoom**(GSAGE-GCN, $l$=1) | **0.621** | 4.8 (2.0×) | **0.923** | 3.4 (3.0×) |
| **GZoom**(GSAGE-GCN, $l$=2) | 0.612 | **1.8 (5.2×)** | 0.917 | **1.6 (6.3×)** |
| GraphSAGE-mean | 0.598 | 11.1 | 0.897 | 8.1 |
| **GZoom**(GSAGE-mean, $l$=1) | 0.614 | 5.2 (2.2×) | **0.925** | 2.6 (3.1×) |
| **GZoom**(GSAGE-mean, $l$=2) | **0.617** | **1.8 (6.2×)** | 0.919 | **1.2 (6.8×)** |
| GraphSAGE-LSTM | 0.596 | 387.3 | 0.907 | 92.2 |
| **GZoom**(GSAGE-LSTM, $l$=1) | 0.614 | 151.8 (2.6×) | **0.920** | 39.8 (2.3×) |
| **GZoom**(GSAGE-LSTM, $l$=2) | **0.615** | **52.5 (7.4×)** | 0.917 | **14.5 (6.4×)** |
| GraphSAGE-pool | 0.602 | 144.9 | 0.892 | 84.3 |
| **GZoom**(GSAGE-pool, $l$=1) | 0.611 | 66.0 (2.2×) | **0.921** | 27.0 (3.1×) |
| **GZoom**(GSAGE-pool, $l$=2) | **0.614** | **23.4 (6.2×)** | 0.912 | **12.4 (6.8×)** |

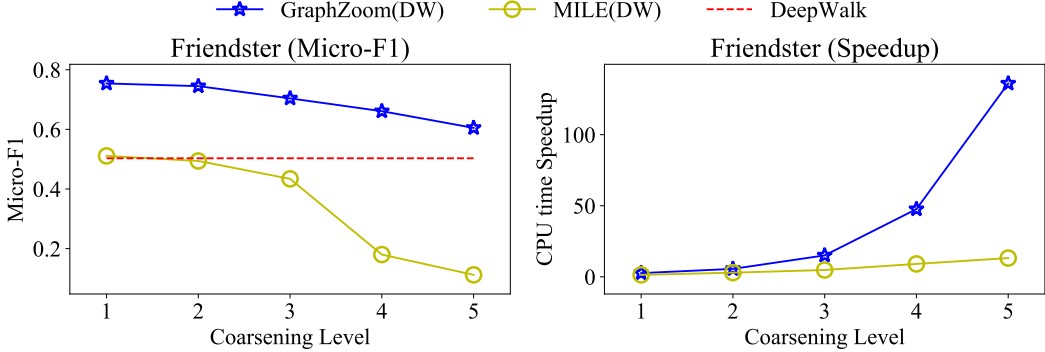

Figure 2: Comparisons of GraphZoom and MILE on Friendster dataset.

we use DeepWalk as the embedding kernel. As shown in Figure 2, GraphZoom drastically boosts the Micro-F1 score up to $47.6\%$ compared to MILE and $49.9\%$ compared to DeepWalk with a speedup up to $119.8\times$. When increasing the coarsening level, GraphZoom achieves a higher speedup while the embedding accuracy decreases gracefully. This shows the key strength of GraphZoom: it can effectively coarsen a large graph by merging many redundant nodes that are spectrally similar, thus preserving the graph spectral (structural) properties that are important to the underlying embedding model. When applying basic embedding model on coarsest graph, it can learn more global information from spectral domain, leading to high-quality node embeddings. On the contrary, heuristic graph coarsening algorithm used in MILE fails to preserve a meaningful coarsest graph, especially when coarsening graph by a large reduction ratio.

### 4.3 ABLATION ANALYSIS ON GRAPHZOOM KERNELS

To study the effectiveness of our proposed GraphZoom kernels separately, we compare each of them against the corresponding kernel in MILE while fixing other kernels. As shown in Figure 3, when fixing coarsening kernel and comparing refinement kernel of GraphZoom with that of MILE, Our re-

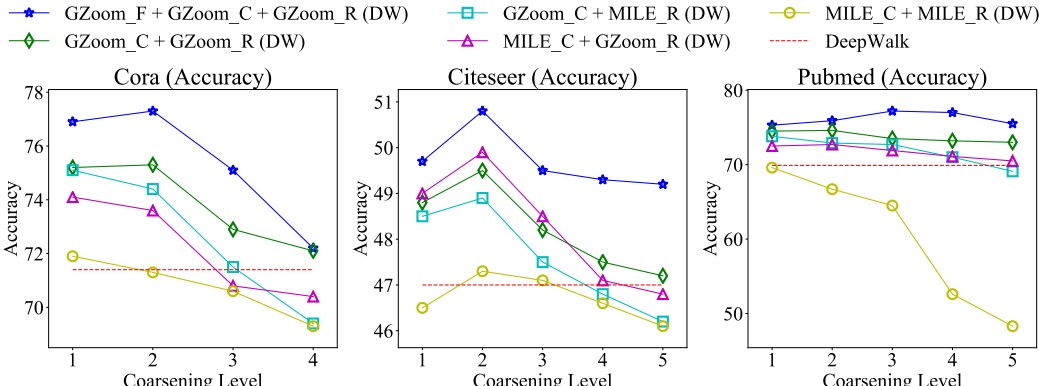

Figure 3: Comparisons of different kernel combinations in GraphZoom and MILE in classification accuracy on Cora, Citeseer, and Pubmed datasets — We choose DeepWalk (DW) as the embedding kernel. GZoom_F, GZoom_C, GZoom_R denote the fusion, coarsening, and refinement kernels proposed in GraphZoom, respectively; MILE_C and MILE_R denote the coarsening and refinement kernels in MILE, respectively; The blue curve is basically GraphZoom and the yellow one is MILE.

finement kernel can improve embedding results upon MILE refinement kernel, especially when the coarsening level is large. This indicates that our proposed graph filter in refinement kernel can successfully filter out high-frequency noise from the graph to improve embedding quality. Similarly, when comparing coarsening kernels in GraphZoom and MILE with the refinement kernel fixed, GraphZoom coarsening kernel can also improve embedding quality upon MILE coarsening kernel, which shows that our spectral graph coarsening algorithm can indeed retain key graph structure for underlying graph embedding models to exploit. When combining the GraphZoom coarsening and refinement kernels, we can achieve a better classification accuracy compared to the ones using any other kernels in MILE. This suggests that the GraphZoom coarsening and refinement kernels play useful yet distinct roles to boost embedding performance and their combination can further improve embedding result. Moreover, adding graph fusion improves classification accuracy by a large margin, which indicates that graph fusion can properly incorporate both graph topology and node attribute information that are crucial for lifting the embedding quality of downstream embedding models. Results of each kernel CPU time and speedup comparison are available in Appendix G and Appendix I.

## 5   CONCLUSIONS

This work introduces GraphZoom, a multi-level framework to improve the accuracy and scalability of unsupervised graph embedding tasks. GraphZoom first fuses the node attributes and topology of the original graph to construct a new weighted graph. It then employs spectral coarsening to generate a hierarchy of coarsened graphs, where embedding is performed on the smallest graph at the coarsest level. Afterwards, proper graph filters are used to iteratively refine the graph embeddings to obtain the final result. Experiments show that GraphZoom improves both classification accuracy and embedding speed on a number of popular datasets. An interesting direction for future work is to derive a proper way to propagate node labels to the coarsest graph, which would allow GraphZoom to support supervised graph embedding.

ACKNOWLEDGMENTS

This work is supported in part by Semiconductor Research Corporation and DARPA, Intel Corporation under the ISRA Program, and NSF Grants #1350206, #1909105, and #1618364. We would like to thank Prof. Zhengfu Xu of Michigan Technological University for his helpful discussion on the formulation of the embedding refinement problem.

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

## APPENDIX A    DETAILS OF DATASETS

**Transductive task.**    We follow the experiments setup in Yang et al. (2016) for three standard citation network benchmark datasets: Cora, Citeseer, and Pubmed. In all these three citation networks, nodes represent documents and edges correspond to citations. Each node has a sparse bag-of-word feature vector and a class label. We allow only 20 labels per class for training and $1,000$ labeled nodes for testing. In addition, we further evaluate on Friendster dataset (Yang & Leskovec, 2015), which contains 8 million nodes and 400 million edges, with $2.5\%$ of the nodes used for training and $0.3\%$ nodes for testing. In Friendster, nodes represent users and a pair of nodes are linked if they are friends; each node has a class label but is not associated with a feature vector.

**Inductive task.**    We follow Hamilton et al. (2017) for setting up experiments on both protein-protein interaction (PPI) and Reddit dataset. PPI dataset consists of graphs corresponding to human tissues, where nodes are proteins and edges represent interaction effects between proteins. Reddit dataset contains nodes corresponding to users' posts: two nodes are connected through an edge if the same users comment on both posts. We use $60\%$ nodes for training, $40\%$ for testing on PPI and $65\%$ for training and $35\%$ for testing on Reddit.

## APPENDIX B    DETAILS OF BASELINES

**DeepWalk** first generates random walks based on graph structure. Then, walks are treated as sentences in a language model and Skip-Gram model is exploited to obtain node embeddings.

**node2vec** is different from DeepWalk in terms of generating random walks by introducing the return parameter $p$ and the in-out parameter $q$, which can combine DFS-like and BFS-like neighborhood exploration.

**Deep Graph Infomax (DGI)** is an unsupervised approach that generates node embeddings by maximizing mutual information between patch representations (local information) and corresponding high-level summaries (global information) of graphs.

**GraphSAGE** embeds nodes in an inductive way by learning an aggregation function that aggregates node features to obtain embeddings. GraphSAGE supports four different aggregation functions: GraphSAGE-GCN, GraphSAGE-mean, GraphSAGE-LSTM and GraphSAGE-pool.

**HARP** coarsens the original graph into several levels and apply underlying embedding model to train the coarsened graph at each level sequentially to obtain the final embeddings on original graph. Since the coarsening level is fixed in their implementation, we run HARP in our experiments without changing the coarsening level.

**MILE** is the state-of-the-art multi-level unsupervised graph embedding framework and similar to our GraphZoom framework since it also contains graph coarsening and embedding refinement kernels. More specifically, MILE first uses its heuristic-based coarsening kernel to reduce the graph size and trains underlying unsupervised graph embedding model on coarsest graph. Then, its refinement kernel employs Graph Convolutional Network (GCN) to refine embeddings on the original graph. We compare GraphZoom with MILE on various datasets, including Friendster that contains 8 million nodes and 400 million edges (shown in Table 2 and Figure 2). Moreover, we further compare each kernel in GraphZoom and MILE in Figure 3.

## APPENDIX C    INTEGRATE GRAPHZOOM WITH GRAPHSAGE AND DGI

As both GraphSAGE and DGI require node features for embedding on coarsest graph, we map the initial node feature matrix $X_0$ to the coarsened graph by iteratively performing $X_{i+1} = \hat{H}_i^{i+1} X_i$, where $\hat{H}_i^{i+1}$ is the mapping operator $H_i^{i+1}$ with $l^1$-normalization per row. When applying DGI to the coarsest graph (i.e., the $l$-th coarsening level), the node embeddings $E_l$ may lose local information due to the property of DGI. Inspired by skip connection (He et al., 2016) and Graph U-nets (Gao & Ji, 2019), we use $\hat{E}_l = E_l || X_l$ as the node embeddings on the coarsest graph, where $||$ denotes featurewise concatenation.

## APPENDIX D   GRAPH SIZE AT DIFFERENT COARSENING LEVEL

Table 4: Number of nodes at different GraphZoom coarsening levels. GZoom-0 means GraphZoom with 0 coarsening level (i.e., without coarsening), GZoom-1 means GraphZoom with 1 coarsening level and so forth. "—" means number of nodes is less than 50.

| Dataset | GZoom-0 | GZoom-1 | GZoom-2 | GZoom-3 | GZoom-4 | GZoom-5 |
|---|---|---|---|---|---|---|
| Cora | 2,708 | 1,169 | 519 | 218 | 100 | — |
| Citeseer | 3,327 | 1,488 | 606 | 282 | 131 | 58 |
| Pubmed | 19,717 | 7,903 | 3,562 | 1,651 | 726 | 327 |
| PPI | 14,755 | 5,061 | 1,815 | 685 | 281 | 120 |
| Reddit | 232,965 | 84,562 | 30,738 | 11,598 | 4,757 | 2,117 |
| Friendster | 7,944,949 | 2,734,483 | 1,048,288 | 409,613 | 134,956 | 44,670 |

## APPENDIX E   GRAPHZOOM ALGORITHM

---

**Algorithm 1:** GraphZoom algorithm

---

   **Input:** Adjacency matrix $\boldsymbol{A}_{topo} \in \mathbb{R}^{N \times N}$; node feature matrix $\boldsymbol{X} \in \mathbb{R}^{N \times K}$;
        base embedding function $g(\cdot)$; coarsening level $l$
   **Output:** Node embedding matrix $\boldsymbol{E} \in \mathbb{R}^{N \times D}$

1  $\boldsymbol{A}_0 = graph\_fusion(\boldsymbol{A}_{topo}, \boldsymbol{X})$;
2  **for** $i = 1...l$ **do**
3     $\boldsymbol{A}_i, \boldsymbol{H}_{i-1}^i = spectral\_coarsening(\boldsymbol{A}_{i-1})$;
4  **end**
5  $\boldsymbol{E}_L = g(\boldsymbol{A}_l)$;
6  **for** $i = l...1$ **do**
7     $\hat{\boldsymbol{E}}_{i-1} = (\boldsymbol{H}_{i-1}^i)^T \boldsymbol{E}_i$;
8     $\boldsymbol{E}_{i-1} = refinement(\hat{\boldsymbol{E}}_{i-1})$;
9  **end**
10 $\boldsymbol{E} = \boldsymbol{E}_0$;

---

## APPENDIX F   SPECTRAL COARSENING

Coarsening is one type of graph reduction whose objective is to achieve computational acceleration by reducing the size (i.e., number of nodes) of the original graphs while maintaining the similar graph structure between the original graph and the reduced ones. For each coarsen level $i + 1$, a surjection is defined between the original node set $\mathcal{V}_i$ and the reduced node set $\mathcal{V}_{i+1}$, where each node $v \in \mathcal{V}_{i+1}$ corresponds to a small set of adjacent vertices in the original node set $\mathcal{V}_i$. Mapping operator $H_i^{i+1}$ can be defined based on the mapping relationship between $\mathcal{V}_i$ and $\mathcal{V}_{i+1}$, Note that the operator $H_i^{i+1} \in \{0,1\}^{|\mathcal{V}_{i+1}| \times |\mathcal{V}_i|}$ is a matrix containing only 0s and 1s. It has following properties:

- The row (column) index of $H_i^{i+1}$ corresponds to the node index in graph $\mathcal{G}_{i+1}$ ($\mathcal{G}_i$).

- It is a surjective mapping of the node set, where $(H_i^{i+1})_{p,q} = 1$ if node $q$ in graph $\mathcal{G}_i$ is aggregated to super-node $p$ in graph $\mathcal{G}_{i+1}$, and $(H_i^{i+1})_{p',q} = 0$ for all nodes $p' \in \{v \in \mathcal{V}_{i+1} : v \neq p\}$.

- It is a locality-preserving operator, where the coarsened version of $\mathcal{G}_i$ induced by the non-zero entries of $(H_i^{i+1})_{p,:}$ is connected for each $p \in \mathcal{V}_{i+1}$.

---

**Algorithm 2:** spectral_coarsening algorithm

---

    **Input:** Adjacency matrix $\boldsymbol{A}_i \in \mathbb{R}^{|\mathcal{V}_i| \times |\mathcal{V}_i|}$

    **Output:** Adjacency matrix $\boldsymbol{A}_{i+1} \in \mathbb{R}^{|\mathcal{V}_{i+1}| \times |\mathcal{V}_{i+1}|}$ of the reduced graph

                  $\mathcal{G}_{i+1}$, mapping operator $\boldsymbol{H}_i^{i+1} \in \mathbb{R}^{|\mathcal{V}_{i+1}| \times |\mathcal{V}_i|}$

**1** $n = |\mathcal{V}_i|$, $n_c = n$;

**2** [graph reduction ratio] $\gamma_{max} = 1.8$ , $\delta = 0.9$;

**3 for** *each edge* $(p, q) \in \mathcal{E}_i$ **do**

**4**    | [spectral node affinity set] $\mathbb{C} \leftarrow a_{p,q}$ defined in Eq. 2 ;

**5 end**

**6 for** *each node* $p \in \mathcal{V}_i$ **do**

**7**    | $d(p) = \left|(A_i)_{p,:}\right|$, $d_m(p) = \text{median}\left(\left|(A_i)_{q,:}\right| \text{ for all } q \in \{q|(p,q) \in \mathcal{E}_i\}\right)$;

**8**    | **if** $d(p) \geq 8 \cdot d_m(p)$ **then**

**9**       | [node aggregation flag] $\boldsymbol{z}(p) = 0$;

**10**    | **else**

**11**       | [node aggregation flag] $\boldsymbol{z}(p) = -1$;

**12**    | **end**

**13 end**

**14** $\gamma = 1$;

**15 while** $\gamma < \gamma_{max}$ **do**

**16**    | $\mathbb{S} = \emptyset$ , $\mathbb{U} = \emptyset$ ;

**17**    | [unaggregated node set] $\mathbb{U} \leftarrow p \in \{p|z(p) == -1 \; \forall p \in \mathcal{V}_i\}$;

**18**    | $\mathbb{S} \leftarrow p \in \{p|a_{p,q} \geq \delta \cdot \max\limits_{s \neq p,q}\left(\max\limits_{(p,s)\in\mathcal{E}_i}(a_{p,s}), \max\limits_{(q,s)\in\mathcal{E}_i}(a_{q,s})\right) \; \forall p \in \mathcal{V}_i\}$;

**19**    | **for** *each node* $p$ *in* $\mathbb{S} \cap \mathbb{U}$ **do**

**20**       | **if** $\boldsymbol{z}(p) == -1$ **then**

**21**          | $q = \underset{(p,q)\in\mathcal{E}_i}{\arg\max}\left(a_{p,q}\right)$;

**22**

**23**          | **if** $\boldsymbol{z}(q) == -1$ **then**

**24**            | $\boldsymbol{z}(q) = 0, \boldsymbol{z}(p) = q, \hat{q} = q$;

**25**          | **else if** $\boldsymbol{z}(q) == 0$ **then**

**26**            | $\boldsymbol{z}(p) = q, \hat{q} = q$;

**27**          | **else**

**28**            | $\boldsymbol{z}(p) = \boldsymbol{z}(q), \hat{q} = \boldsymbol{z}(q)$;

**29**          | **end**

**30**          | update smoothed vectors in $T$ with $x_p^{(k)} = x_{\hat{q}}^{(k)}$ for $k = 1, \cdots, t$

                $n_c = n_c - 1$;

**31**       | **end**

**32**    | **end**

**33**    | $\gamma = n/n_c, \delta = 0.7 \cdot \delta$ ;

**34**    | $\boldsymbol{z}(p) = p$ for node $p \in \{p|\boldsymbol{z}(p) == 0\}$

**35 end**

**36** form $\boldsymbol{H}_i^{i+1}$ and $\boldsymbol{A}_{i+1}$ based on $\boldsymbol{z}$

---

## APPENDIX G    CPU TIME OF EACH GRAPHZOOM KERNEL

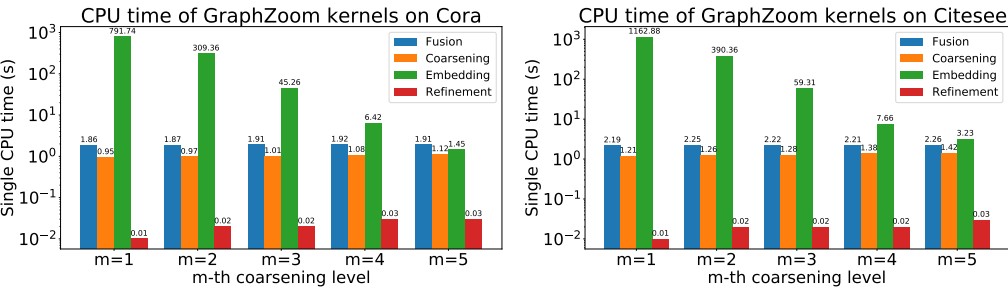

(a) GraphZoom with DeepWalk as embedding kernel    (b) GraphZoom with DeepWalk as embedding kernel

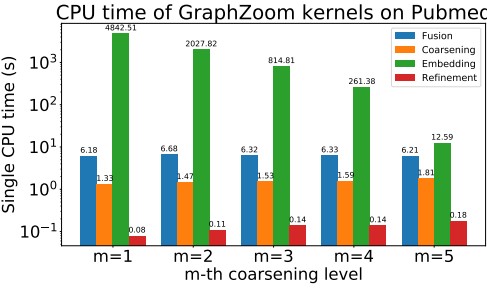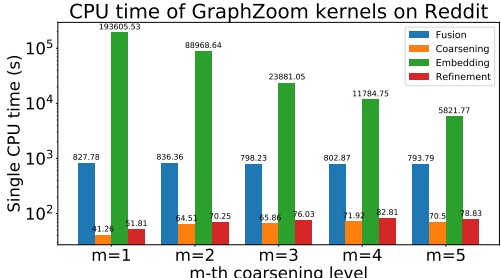

(c) GraphZoom with DeepWalk as embedding kernel    (d) GraphZoom with GSAGE as embedding kernel

Figure 4: CPU time of GraphZoom kernels

As shown in Figure 4 (note that the y axis is in logarithmic scale), the GraphZoom embedding kernel dominates the total CPU time, which can be more effectively reduced with a greater coarsening level $L$. All other kernels in GraphZoom are very efficient, which enable the GraphZoom framework to drastically reduce the total graph embedding time.

## APPENDIX H    GRAPH FILTERS AND LAPLACIAN EIGENVALUES

Figure 5a shows the original distribution of graph Laplacian eigenvalues which also can be interpreted as frequencies in graph spectral domain (smaller eigenvalue means lower frequency). The proposed graph filter for embedding refinement (as shown in Figure 5e) can be considered as a band-stop filter that passes all frequencies with the exception of those within the middle stop band that is greatly attenuated. Therefore, the band-stop filter may not be very effective for removing high-frequency noises from the graph signals. Fortunately, it has been shown that by adding self-loops to each node in the graph as follows $\tilde{A} = A + \sigma I$ (shown in Figure 5b, 5c, 5d, where $\sigma = 0.5, 1.0, 2.0$), the distribution of Laplacian eigenvalues can be squeezed to the left (towards zero) (Maehara, 2019). By properly choosing $\sigma$ such that large eigenvalues will mostly lie in the stop band (e.g., $\sigma = 1.0, 2.0$ shown in Figure 5c and 5d), the graph filter will be able to effectively filtered out high-frequency components (corresponding to high eigenvalues) while retaining low-frequency components, which is similar to a low-pass graph filter as shown in Figure 5f. It is worth noting that if $\sigma$ is too large, then most eigenvalues will be very close to zero, which makes the graph filter less effective for removing noises. In this work, we choose $\sigma = 2.0$ for all our experiments.

## APPENDIX I    SPEEDUP OF GRAPHZOOM KERNELS COMPARED TO MILE

As shown in Figure 6, the combination of GraphZoom coarsening and refinement kernels can always achieve the greatest speedups (green curves); adding GraphZoom fusion kernel (blue curves) will lower the speedups by a small margin but further boost the embedding quality, showing a clear trade-

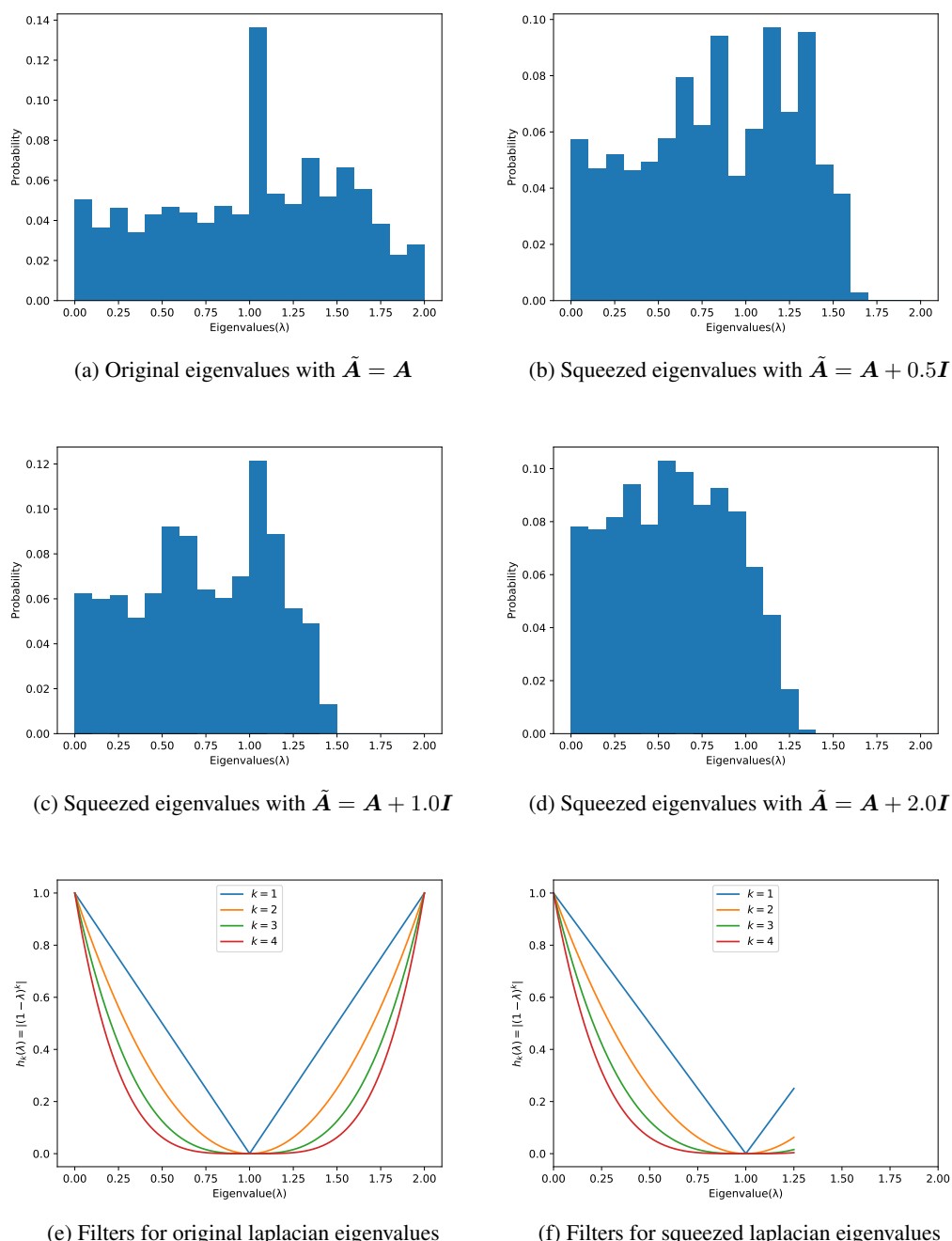

(a) Original eigenvalues with $\tilde{A} = A$

(b) Squeezed eigenvalues with $\tilde{A} = A + 0.5I$

(c) Squeezed eigenvalues with $\tilde{A} = A + 1.0I$

(d) Squeezed eigenvalues with $\tilde{A} = A + 2.0I$

(e) Filters for original laplacian eigenvalues

(f) Filters for squeezed laplacian eigenvalues

Figure 5: Distribution of graph laplacian eigenvalues with different self-loops on Cora

off between embedding quality and runtime efficiency: to achieve the highest graph embedding quality, the graph fusion kernel should be included.

## APPENDIX J  MORE RESULTS ON PPI DATASETS

As shown in Figure 7, the embedding results by GraphZoom with increasing number of coarsening levels are still better than the ones by GraphSAGE with different aggregation functions. It is worth noting that although the level-5 (coarsened) graph contains only 120 nodes (as shown in D), Graph-

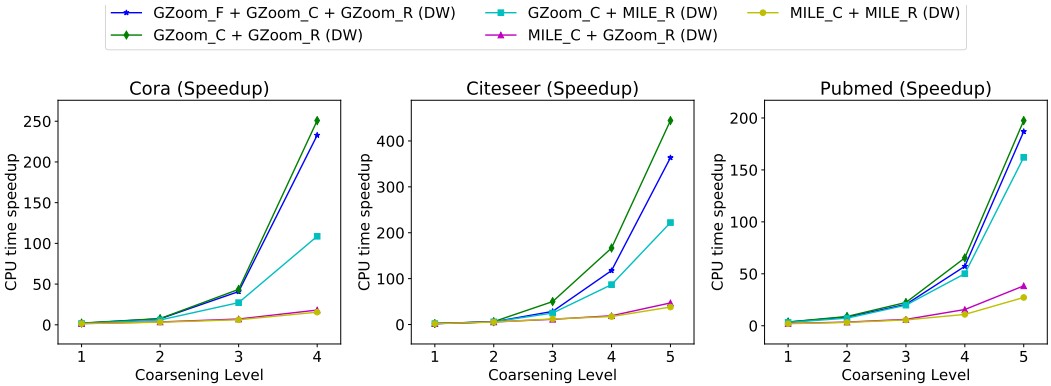

Figure 6: Comparisons of different combinations of kernels in GraphZoom and MILE in terms of CPU time speedup on Cora, Citeseer, and Pubmed datasets — We choose DeepWalk (DW) as basic embedding method. GZoom_F, GZoom_C, GZoom_R, MILE_C and MILE_R represent GraphZoom fusion kernel, GraphZoom coarsening kernel, GraphZoom refinement kernel, MILE coarsening kernel and MILE refinement kernel, respectively.

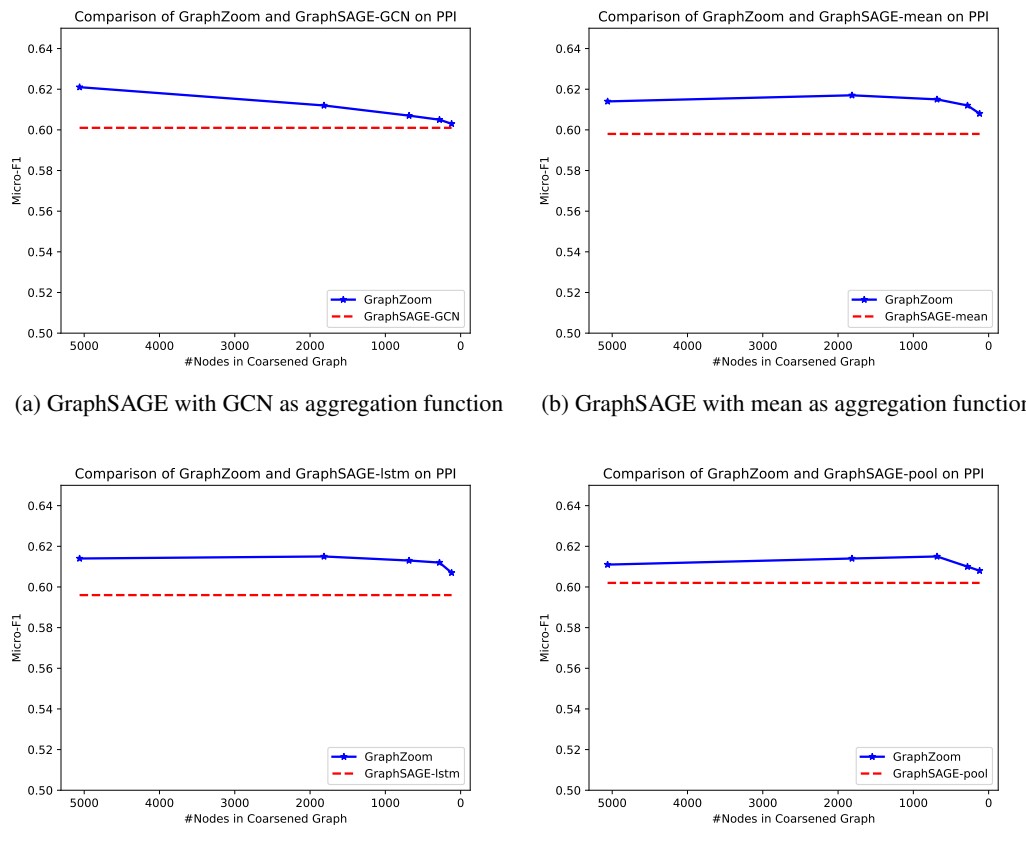

(a) GraphSAGE with GCN as aggregation function     (b) GraphSAGE with mean as aggregation function

(c) GraphSAGE with LSTM as aggregation function     (d) GraphSAGE with pool as aggregation function

Figure 7: Comparisons of GraphZoom and GraphSAGE on PPI

Zoom embedding results can still beat the GraphSAGE baseline obtained with the original graph containing $14,755$ nodes.

## APPENDIX K   MORE RESULTS ON NON-ATTRIBUTED DATASETS

Table 5: Node classification results on PPI and Wiki datasets.

| Method | PPI(Homo Sapiens) | | Wiki | |
|---|---|---|---|---|
| | Mircro-F1 | Time(mins) | Mircro-F1 | Time(s) |
| DeepWalk | 0.231 | 2.3 | 0.705 | 94.4 |
| MILE(DW, $l$=1) | 0.256 | 1.1 (2.1×) | 0.703 | 42.5 (2.2×) |
| MILE(DW, $l$=2) | 0.253 | 0.7 (3.3×) | 0.639 | 20.9 (4.5×) |
| **GZoom**(DW, $l$=1) | **0.261** | 0.6 (3.8×) | **0.726** | 29.4 (3.2×) |
| **GZoom**(DW, $l$=2) | 0.255 | **0.2 (11.5×)** | 0.678 | **6.7 (14.1×)** |

Table 6: Link prediction results on PPI and Wiki datasets.

| Method | PPI(Homo Sapiens) | | Wiki | |
|---|---|---|---|---|
| | AUC | Time(mins) | AUC | Time(mins) |
| DeepWalk | 0.721 | 5.2 | 0.774 | 3.0 |
| MILE(DW, $l$=1) | 0.772 | 3.7 (1.4×) | 0.877 | 1.6 (1.8×) |
| MILE(DW, $l$=2) | 0.701 | 2.3 (2.3×) | 0.868 | 0.9 (3.3×) |
| MILE(DW, $l$=3) | 0.723 | 1.2 (4.3×) | 0.842 | 0.5 (6.0×) |
| **GZoom**(DW, $l$=1) | 0.818 | 2.1 (2.5×) | 0.901 | 1.2 (2.5×) |
| **GZoom**(DW, $l$=2) | 0.834 | 0.7 (7.4×) | 0.902 | 0.4 (7.5×) |
| **GZoom**(DW, $l$=3) | **0.855** | **0.1 (52.0×)** | **0.908** | **0.1 (30.0×)** |

To further show that GraphZoom can work on non-attributed datasets, we evaluate it on PPI(Homo Sapiens) and Wiki datasets, following the same dataset configuration as used in Grover & Leskovec (2016); Liang et al. (2018). As shown in Table 5, GraphZoom (without fusion kernel) improves the performance of basic embedding model (i.e., DeepWalk) at the first coarsening level. When further increasing coarsening level, GraphZoom achieves much larger speedup while its performance may drop a little bit. For link prediction task, we follow Grover & Leskovec (2016) by choosing Hadamard operator to transform node embeddings into edge embeddings. Our link prediction results show that GraphZoom significantly improve performance by a margin of $18.5\%$ with speedup up to $52.0\times$, showing GraphZoom can generate high-quality embeddings for various tasks.

## APPENDIX L   SENSITIVITY ANALYSIS OF GRAPH FUSION KERNEL

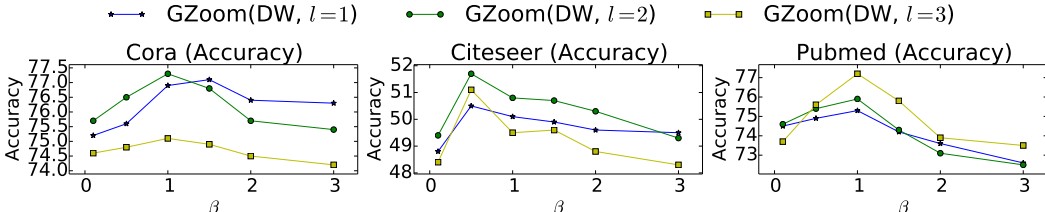

Figure 8: Comparisons of classification accuracy using various $\beta$ values in the graph fusion kernel for the Cora, Citeseer, and Pubmed datasets. $l$ represents the $l$-th coarsening level.

As graph fusion kernel fuses node attribute with graph topology via $A_{fusion} = A_{topo} + \beta A_{feat}$, the value of $\beta$ plays a critical role in balancing attribute and topological information. On the one hand, if $\beta$ is too small, we may lose too much node attribute information. On the other hand, if $\beta$ is too

large, the node attribute information will dominate the fused graph and therefore graph topological information is undermined. As shown in Figure 8, we vary $\beta$ from 0.1 to 3 in graph fusion kernel and evaluate it on Cora, Citeseer, and Pubmed datasets. The results indicate that $\beta = 1$ achieves the best performance on Cora and Pubmed, while $\beta = 0.5$ is the best configuration on Citeseer. We choose $\beta = 1$ in all our experiments.

