# OpenReview forum: "GraphZoom: A Multi-level Spectral Approach for Accurate and Scalable Graph Embedding"
_ICLR.cc/2020/Conference — Accept (Talk)_

### Official Review · AnonReviewer2 · 2019-10-23
**Official Blind Review #2**

**Rating:** 6

**Review:**

Summary: The authors propose a way to fuse information on nodes of a graph with the topology of the graph in the large scale setting. The proposed approach is done in four phases where (i) the covariates in the nodes of the graph is first mapped in the graph space for fusion and fused using linear combination of the topological graph and feature graph, (ii) the resulting "adjacency" matrix will almost surely not be sparse even if the original graph space, so they use eigenvalues of the graph laplacian to coarsen the graph -- remove edges; (iii) they then propose to embed the coarsened graph using "any" unsupervised learning technique; (iv) then the embedded representation is refined using iterative procedures. Cheap procedures are introduced to do Phases (i) and (iv). Experimentally the authors see improvements in the performance using their approach compared to the baselines considered.

Novelty: 1. The approach suggested in this paper is already there in MILE Fig 1., the authors mention that MILE requires training GCN but I am not sure why this is critical. The authors mention that "MILE cannot support inductive embedding
models due to the transductive property of GCN", can you clarify what this means? I guess one can easily replace GCNs?

2. Covariate adjusted clustering is known to work only when when the features are independent like Stochastic Block Model, see  Covariate-assisted spectral clustering by Binkiewicz et al, 2014. Is there a reason why the features that we see on nodes are not correlated?

Results: It is hard to see where the performance improvement actually comes from, if at all. It is interesting to see that the proposed approach saves time and is more accurate in the variety of settings considered, but it is not clear why we see the improvement.

After rebuttal: I have raised my score to 6 after going through the authors' response for my questions, and other reviewers' concerns. While the approach performs well in many datasets (thanks to the authors for providing more experimental evidence!), I'm still not convinced with the authors' response on their fusion step -- it seems to me that node attributes are "side" information, that can "boost" the signal on the original neighborhood graph. Recall that spectral approaches do have a fundamental barrier -- they fail on "thin" graphs (see https://arxiv.org/abs/1608.04845 ). Hence,  node covariance/fusion matrix being dense will be a blessing for spectral approaches since they make spectral methods work. However, is this what we want in *all* the cases? I'm not sure. This means that the choice of \beta in their fusion step is *very* important, and I don't see any plots on the sensitivity of their procedure with respect to \beta. I kindly request the authors to include a plot or results showing the sensitivity of the final results with respect to the choice of \beta. Thanks!

**Experience Assessment:**

I have read many papers in this area.

**Review Assessment: Checking Correctness Of Derivations And Theory:**

I carefully checked the derivations and theory.

**Review Assessment: Checking Correctness Of Experiments:**

I assessed the sensibility of the experiments.

**Review Assessment: Thoroughness In Paper Reading:**

I read the paper thoroughly.

---

> ### Author Response · Authors · 2019-11-10
> **Additional Clarifications**
>
> Thanks for your review. We are going to address your concerns one by one as follows:
>
> Q1: The approach suggested in this paper is already there in MILE Fig 1.
>
> A1: Figure1 in MILE basically shows multi-level optimization, which is a very general approach and its application to graph embedding is not an invention of MILE. There are some other multi-level graph embedding techniques discussed in our related work. Although both GraphZoom and MILE can be viewed as multi-level optimization methods, they are derived in totally different ways. Quite different from MILE, GraphZoom is the first multi-level **spectral** approach to graph embedding. More specifically, our work proposes very efficient and effective coarsening & refinement algorithms based on spectral graph theory, which can guarantee that the first few eigenvectors of graph Laplacian matrix can be preserved while high-frequency noise is filtered out. More importantly, GraphZoom has an additional graph fusion kernel to further improve embedding quality, which does not exist in MILE paper.
>
> Q2: Why not replacing GCN to make MILE support inductive task? Why training GCN in MILE is critical?
>
> A2: As shown in MILE paper, the authors try to exploit inductive models (e.g., GraphSAGE) but find it is not as good as GCN. The author claims that training GCN model in refinement stage achieves the best performance. However, it makes MILE only applicable to transductive learning, since it requires knowing the whole graph structure information to train GCN model during its refinement stage. Moreover, training GCN even on coarest graph is very time consuming ( 5~6 hours on graphs with 100k nodes), which limits the speedup of MILE over basic embedding models. As shown in Table 2, Table 3, Figure 2, and Appendix H, MILE achieves much smaller speedup than GraphZoom, showing the major drawback of training GCN.
>
> Q3: Covariate adjusted clustering is known to work only when the features are independent like Stochastic Block Model, see  Covariate-assisted spectral clustering by Binkiewicz et al, 2014. Is there a reason why the features that we see on nodes are not correlated?
>
> A3: Thanks for the question. Our method is fundamentally different from the covariate-assisted spectral clustering method. The ideal fusion step would require constructing the nearest-neighbor (NN) graph using the complete set of node features, while the similarities between node features will be encoded using edge weights: two nodes’ features with higher similarities will lead to a greater edge weight. To gain higher efficiency, instead of constructing NN graphs using all node features, we propose to construct local NN graphs for each node cluster (identified through spectral coarsening of the topology graph). In this way, we are able to leverage graph topological properties for constructing approximate NN graphs in linear time. As a result, when nodes are properly ordered, the Laplacians of the NN graphs constructed using our approach will be sparse block-diagonal matrices. The sparsity of the Laplacian matrix will depend on the number of the nearest neighbors for constructing the NN graph. By combining the topology graph Laplacian with the NN graph Laplacian in the graph fusion step, we are able to further improve the quality of the proposed spectral coarsening procedure: a key step that determines how the nodes on each level should be aggregated for forming the coarser graph without impacting the overall embedding accuracy.
>
> On the other hand, the covariate-assisted spectral clustering method directly uses the covariance matrix constructed using node features. The resultant matrix can be very dense in general and will not be a graph Laplacian matrix. We will further clarify these differences in the future revision.
>
> Q4: It is hard to see where the performance improvement actually comes from, if at all. It is interesting to see that the proposed approach saves time and is more accurate in the variety of settings considered, but it is not clear why we see the improvement.

---

> > ### Author Response · Authors · 2019-11-10
> > **Continued anwser**
> >
> > A4: We have discussed in our paper that graph fusion kernel can enable underlying graph embedding method incorporate node attribute information; In addition, both graph coarsening and embedding refinement kernels effectively preserve the first few eigenvectors of the Laplacian matrix and remove high frequency noise from the graph. Reducing such noise can significantly improve embedding quality as noted by [Maehara, Takanori. "Revisiting Graph Neural Networks: All We Have is Low-Pass Filters." arXiv:1905.09550 (2019).], [Li, Qimai, et al. "Label efficient semi-supervised learning via graph filtering." CVPR 2019.]. Moreover, the time complexity of all these three kernels is linear, which explains why GraphZoom is very efficient and can achieve a large speedup compared to basic embedding models. As shown in Figure 3 and Appendix H, we compare each kernel of GraphZoom with the corresponding kernel in MILE to see whether each kernel can improve accuracy and increase speedup compared to MILE.. Results show that each kernel of GraphZoom can individually improve embedding quality and their combination can further increase the accuracy by a large margin.

---

### Official Review · AnonReviewer1 · 2019-10-25
**Official Blind Review #1**

**Rating:** 8

**Review:**

Summary: This paper proposes GraphZoom, a framework for augmenting unsupervised graph embedding methods by (a) fusing feature information into the graph topology, (b) learning embeddings on a coarsened graph, and (c) refining the coarsened embeddings to obtain embeddings for the original graph nodes. In particular, a nearest neighbor graph over node features is computed and this adjacency matrix is linearly combined with the original adjacency matrix to obtain a graph with feature information "fused in". The graph is then coarsened using a spectral approach, embeddings are learned on the coarsened graph (via any strategy), and the embeddings are then refined back to the original nodes (again using a spectral approach). The authors take care to heed the advice of Maehara et al. and remove high-frequency information from the features.

Assessment: Overall, this is a borderline contribution with some interesting motivation, original ideas, and sound derivations. However, the primary limitation of this work is the empirical comparison. First, the empirical comparison includes DeepWalk and GraphSAGE as the two base models, and while these are reasonable models, they are known to no longer be state of the art in this area (e.g., see https://arxiv.org/pdf/1809.10341.pdf). It would be more appropriate to include a more recent and better performing method (e.g., DGI; linked previously), as the reported numbers are very far from state-of-the-art. In addition---and perhaps a more concerning issue---is that seems that a randomly initialized GCN can obtain similar or superior performance compared to the numbers reported in this work (again, see the DGI paper linked above). While it is possible that GraphZoom+DGI or GraphZoom+[some other more recent method] could achieve stronger results, the fact that the current results seem to be below performance of a randomly initialized GCN is a major issue. Stronger empirical results with better baselines and base models would drastically improve the paper.

As another point regarding the empirical results, the datasets used are known to be problematic (e.g., see https://arxiv.org/abs/1811.05868). If these datasets are used, then multiple random splits should be employed and more robust summary statistics should be reported.

Regarding the fusion step, there were also two points that should be addressed in the paper:
1) It seems that this fusion setup is assuming that the network exhibits homophily (i.e., it assumes that nearby nodes have similar features). This is common in many networks (e.g., the benchmarks that are analyzed) but not always the case. Some commentary on when (if ever) this fusion process might *not* be appropriate would improve the paper.
2) The authors state the they use the coarsening process to compute the nearest neighbor graph in order to avoid the quadratic time complexity. However, there are numerous well-established approaches to deal with this issues (e.g., locality sensitive hashing). Why was one of these standard approaches not employed?

Reasons to accept:
- Original and well-motivated idea
- Clearly written paper

Reasons to reject:
- Problematic empirical evaluation (e.g., lacking recent baselines)
- Several performance numbers appear to be below random GCN baseline performance
- General applicability of the approach (e.g., to non-homophilous networks) is not clear

**Experience Assessment:**

I have published in this field for several years.

**Review Assessment: Checking Correctness Of Derivations And Theory:**

I assessed the sensibility of the derivations and theory.

**Review Assessment: Checking Correctness Of Experiments:**

I carefully checked the experiments.

**Review Assessment: Thoroughness In Paper Reading:**

I read the paper at least twice and used my best judgement in assessing the paper.

---

> ### Author Response · Authors · 2019-11-10
> **Additional Experiments and Clarifications**
>
> Thanks for your detailed review. We submitted the revised paper according to your comments. We are going to address your concerns one by one as follows:
>
> Q1: Why not comparing with DGI?
>
> A1: Our paper not only focuses on embedding quality but also scalability for embedding on large graphs (with more than 5 million nodes). However, the datasets with large graphs usually do not have node attribute information (e.g., Friendster). Since DGI requires node attributes for embedding, we did not compare with DGI in our submission. Nevertheless, we agree with you that it is a good idea to include DGI results for other small datasets. So we added the comparison on Cora, Citeseer, and Pubmed in the revised paper (marked in blue). Our results show that GraphZoom+DGI can achieve comparable or better accuracy compared to DGI while achieving speedup up to 11.2x on transductive datasets. Since the authors of DGI have not released their inductive version, we are unable to make quantitative comparison on inductive datasets, although we believe GraphZoom can achieve similar improvements on accuracy and run-time speedup.
>
> Q2: A randomly initialized GCN can obtain similar or superior performance compared to the numbers reported in this work.
>
> A2: Our new results show GraphZoom+DGI can be better than randomly initialized GCN. We think it is not appropriate to compare GraphZoom+DeepWalk with DGI (with randomly initialized GCN encoder) since DeepWalk and DGI are two totally different models. It is worth noting that GraphZoom is a framework rather than an embedding method, which aims to improve the embedding quality of the embedding model plugged into it. In other words, the reason why GraphZoom+DeepWalk is not as good as DGI (with randomly initialized GCN encoder) is basically the fact that GCN based model is superior to DeepWalk on these datasets. A more fair comparison should be comparing GraphZoom+DGI with DGI (or GraphZoom+DeepWalk with DeepWalk), which we are able to show accuracy improvement as well as speedup. We do not claim that GraphZoom can make any unsupervised embedding methods achieve state-of-the-art performance. In general, the stronger that basic model that is  plugged into GraphZoom, the better the resulting performance is. For example, GraphZoom+DGI achieves very promising results in terms of accuracy as well as speedup on those standard benchmark datasets (better than DGI with/without random initialized GCN encoder).
>
> Q3: More robust summary statistics should be reported on these datasets.
>
> A3: Thanks for your suggestion. All of our results (including baselines) are the mean values obtained by running 10 times with a random split training/testing set. The variance is less than 0.3 on transductive datasets and 0.0001 on inductive datasets.
>
> Q4: This fusion process might *not* be appropriate for network that does not exhibit homophily.
>
> A4: This is a very relevant question. In fact, in the fusion step, we do not rely on the assumption that nearby nodes will always have similar features. The ideal fusion step should require constructing the nearest-neighbor (NN) graph using the complete set of node features (the edge weight will encode the similarities between node features), which, however, can be too expensive in practice, even with approximation techniques, such as Locality Sensitive Hashing (LSH). On the other hand, by constructing the NN graph within each node cluster (identified through spectral coarsening of the topology graph), we are able to leverage graph topological properties for constructing approximate NN graphs in linear time, which allows us to further improve the spectral coarsening process. As a result, when nodes are properly ordered, the Laplacians of the NN graphs constructed using our approach will be sparse block-diagonal matrices.
>
> Q5: Why not choosing other well-established approaches (e.g., locality sensitive hashing) to compute the nearest neighbor graph?
>
> A5: Yes, there are many other approaches for approximately constructing nearest-neighbor (NN) graphs, such as the Locality Sensitive Hashing (LSH) method. However, existing methods for constructing NN graphs do not consider graph topological properties when constructing the NN graphs, which may still be not as efficient as our method due to their potentially large constant factors. For example, the LSH-based NN algorithms have a theoretically-low runtime complexity but can even run slower than a linear scan through the data. More detailed discussions about the LSH-based NN graph construction methods can be found in the article "ANN-benchmarks: A benchmarking tool for approximate nearest neighbor algorithms." by Aumüller, Martin et al. (https://arxiv.org/pdf/1807.05614.pdf).

---

### Official Review · AnonReviewer4 · 2019-11-11
**Official Blind Review #4**

**Rating:** 8

**Review:**

The paper provides a multi-level graph-coarsening approach that can improve the predictive and computational performances of numerous existing unsupervised graph embedding models. The proposed approach is a pipeline consisting of 4 steps, viz: 1) Graph Fusion - that fuses attribute similarity graph with network topology, 2> Graph Coarsening - that reduces the graph size iteratively, 3> Graph embedding - using existing models and 4> Embedding refinement. While such a pipeline for scaling using a graph coarsening and refinement based approach is not new, the authors have carefully designed the pipeline to be effective and be scalable such as without any costly learning components (as in mile). The effectiveness of the proposed approach is evaluated with the node classification task on 6 datasets.


Strengths:
- The paper addresses a very important problem. The paper proposes a well-designed pipeline to scale existing embedding models.
- Experimental results support that the proposed approach is effective, especially in terms of reducing computation complexity.

Weaknesses:
- While the experimental results are convincing on the computation front, I have few concerns on the performance front.
   a) 'MILE with the fused graph' baseline is missing. It can been seen from Figure 3 that the incorporation of the attribute graph provides a significant performance benefit. Thus it is necessary to have this baseline to understand the improvement gap w.r.t to MILE. I believe this is a fair comparison to make as the graph fusion component is a commonly used technique in the last decade.
  b) Improvements are inconclusive without additional results on other standard non-attributed graph datasets. In Figure 3, ignoring the model with the fused graph, MILE seems to be comparable to GraphZoom overall. As with the existing results, it's not conclusive whether GraphZoom is better than MILE. Also, add variance and report t-test results.
  c) That said, it can be seen from Figure 2, that GraphZoom significantly outperforms both DW and MILE(DW) on a large non-attributed dataset. However, it is not clear where the significant increase in performance benefits stems from. More analysis is required here.
- Results on other unsupervised embedding task missings. It is important to evaluate the embeddings additionally for the link prediction task at the least.

Additional comment:
- It would be helpful to incorporate one if not some of the attributed graph embedding model as a base model and baseline, such as Deep Graph Infomax (DGI).
- It should be easy to use a mini-batch version of GCN with MILE and use it for inductive learning.
- It would interesting to see what the performance will be without the refinement step.

If my concerns regarding the experiments are positively addressed, I'm willing to improve the score.

-----------------
After the rebuttal, I have updated my score from 3 to 8 as the authors have satisfactorily responded to the concerns raised.



**Experience Assessment:**

I have published one or two papers in this area.

**Review Assessment: Checking Correctness Of Derivations And Theory:**

I assessed the sensibility of the derivations and theory.

**Review Assessment: Checking Correctness Of Experiments:**

I carefully checked the experiments.

**Review Assessment: Thoroughness In Paper Reading:**

I read the paper thoroughly.

---

> ### Author Response · Authors · 2019-11-13
> **Additional Experiments and Clarifications**
>
> Thanks for your detailed review. We submitted the revised paper according to your comments. We are going to address your concerns one by one as follows:
>
> Q1: 'MILE with the fused graph' baseline is missing.
>
> A1: Thanks for your suggestion. We added the comparison in Table 2 (see GZoom_F+MILE). Our results show that adding the fusion kernel will improve MILE performance, though it is still not as good as GraphZoom (with 3-4% accuracy drop).
>
> Q2: Improvements are inconclusive without additional results on other standard non-attributed graph datasets.
>
> A2: As shown in Figure 3, the blue line denotes GraphZoom without the fusion kernel, which shows significant improvement upon MILE (yellow line) on all three transductive datasets (especially on Pubmed). It is worth noting that these transductive datasets can be regarded as non-attributed graphs if you compare blue line with the yellow line in Figure 3 (i.e., we do not consider node attribute information in this case). There are some other papers that treat attributed datasets as non-attributed graphs by only leveraging topological information to graph embedding (e.g., Citeseer dataset in HARP paper) [Chen, Haochen, et al. "Harp: Hierarchical representation learning for networks." AAAI. 2018.]. More importantly, GraphZoom achieves good results on the Friendster dataset, which is non-attributed. To further address your concern, we evaluated GraphZoom on another non-attributed dataset called Deezer(Romania) in Appendix J, following the same training and testing set configuration as GEMSEC paper [Rozemberczki, Benedek, et al. "Gemsec: Graph embedding with self clustering." arXiv:1802.03997 (2018).].
>
> Q3: It can be seen from Figure 2, that GraphZoom significantly outperforms both DW and MILE(DW) on a large non-attributed dataset. However, it is not clear where the significant increase in performance benefits stems from. More analysis is required here.
>
> A3: We agree it is a good idea to include more insights on Friendster results. We **updated** it in the revised paper.
>
> Q4: Results on other unsupervised embedding task missings. It is important to evaluate the embeddings additionally for the link prediction task at the least.
>
> A4: Thanks for the suggestion. Link prediction was not included in our experiments since the baselines we are comparing with (e.g., HARP, MILE, DGI) did not report results for such tasks. Nevertheless, we have evaluated GraphZoom for both transductive and inductive learning tasks, which is consistent with the common practice of measuring the embedding quality for many graph embedding techniques (e.g., GAT, DGI). We plan to experiment with link prediction in our next revision.
>
> Q5: Comparing with DGI?
>
> A5: We totally agree with you that we should compare DGI on these attributed datasets. We added the DGI results in Table 2. The new results (GraphZoom+DGI) achieve promising unsupervised performance and speedups on the standard benchmarks.
>
> Q6: What will GraphZoom performance be without the refinement step?
>
> A6: We added new results of GraphZoom without refinement kernel in Appendix J. Our results indicate that without refinement kernel, the performance will drop 1.5% but still better than DeepWalk and MILE(DW).

---

> > ### Comment · AnonReviewer4 · 2019-11-14
> > **Additional comments based on authors' response**
> >
> > I appreciate that the authors' have responded to all the comments made.  The following are some of my concerns:
> > - It will be helpful to make a complete assessment of the models if results are reported on some of the standard node embedding datasets as used in Node2vec, like BlogCatalog, PPI and wiki.
> > - Also, it would be helpful to report results for link prediction tasks on the above datasets. If not I would ask the authors to explicitly mention that the scope of their model has been tested only for node embeddings.
> > - My primary reason for asking results on these non-attributed graphs is that the datasets used in the paper are highly homophilous (Citation n/ws: CORA, CITESEER, PUBMED)or clusterable (GEMSEC dataset). I would like to see results from less homophilous datasets. Thus, the standard embedding datasets can be a good benchmark.

---

> > > ### Author Response · Authors · 2019-11-15
> > > **Additional Experiments**
> > >
> > > Thanks for your suggestions! Since the rebuttal deadline is approaching, we evaluate GraphZoom on two of the datasets that you suggest (PPI and Wiki) for both node classification and link prediction tasks. As shown in Appendix J, GraphZoom achieves better performance and impressive speedup for both tasks, which confirms that our approach does not rely on homophilous datasets. We hope the new results address your remaining concerns.

---

### Author Response · Authors · 2019-11-15
**Summary of Revision**

Thanks for the comments from all three reviewers. We have added additional experimental results and clarification into our revision (changes marked in blue). Specifically, we added comparison with DGI, GraphZoom_Fusion+MILE, and evaluated GraphZoom on less homophilous datasets for both node classification and link prediction tasks. These experiments provide further evidence that GraphZoom improves the prior arts in both scalability and accuracy. We hope the new empirical results address the main concerns of Reviewer #1 and #4. We also provided more insights on why GraphZoom achieves impressive performance on Friendster and the novelty of our work. We hope Reviewer #2 and #4 would be satisfied with our new clarification.

---

### Decision · Program_Chairs · 2019-12-19

**Decision:**

Accept (Talk)

**Comment:**

The authors present an approach for learning graph embeddings by first fusing the graph to generate a new graph with encodes structural information as well as node attribution information. They then iteratively merge nodes based spectral similarities to  obtain coarser graphs. They then use existing methods to learn embeddings from this coarse graph and progressively refine the embeddings to finer graphs. They demonstrate the performance of their method on standard graph datasets.

This paper has received positive reviews from all reviewers. The authors did a good job of addressing the reviewers' concerns and managed to convince the reviewers about their contributions. I request the authors to take the reviewers suggestions into consideration while preparing the final draft of the paper and recommend that the paper be accepted.